# OMNI-THINKER: SCALING MULTI-TASK RL IN LLMS WITH HYBRID REWARD AND TASK SCHEDULING

**Derek Li**[1][*] **Jiaming Zhou**[1][♠][*] **Leo Maxime Brunswic**[1][*] **Abbas Ghaddar**[1] **Qianyi Sun**[1]
**Liheng Ma**[2] **Yu Luo**[3] **Dong Li**[3] **Mark Coates**[2] **Jianye Hao**[3] **Yingxue Zhang**[1][♠]

[1] Huawei Noah's Ark Lab, Montréal, Canada
[2] McGill University and Mila - Québec AI Institute
[3] Huawei Noah's Ark Lab, Beijing, China

## ABSTRACT

The pursuit of general-purpose artificial intelligence depends on large language models (LLMs) that can handle both structured reasoning and open-ended generation. We present OMNI-THINKER, a unified reinforcement learning (RL) framework that scales LLMs across diverse tasks by combining hybrid rewards with backward-transfer–guided scheduling. Hybrid rewards integrate rule-based verifiable signals with preference-based evaluations from an LLM-as-a-Judge, enabling learning in both deterministic and subjective domains. Our scheduler orders tasks according to accuracy backward transfer (BWT), reducing forgetting and improving multi-task performance. Experiments across four domains show gains of $6.2\%$ over joint training and $12.4\%$ over model merging. Moreover, we demonstrate that simple assumptions on accuracy transfer yield accurate predictions of curriculum outcomes, with entropy dynamics explaining deviations due to generative tasks. These findings underscore the importance of BWT-aware scheduling and hybrid supervision for scaling RL-based post-training toward general-purpose LLMs.

## 1 INTRODUCTION

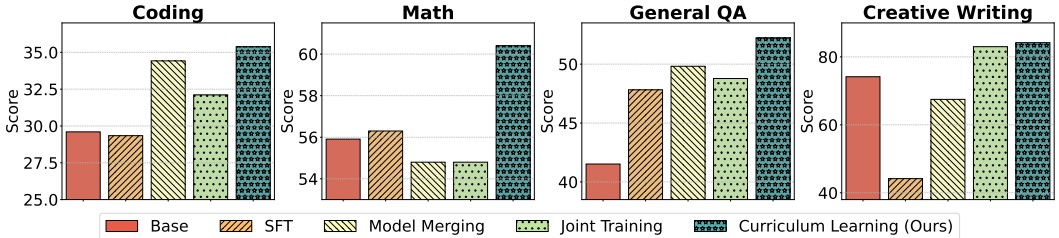

Figure 1: Performance across four task domains, comparing Joint Training and Curriculum Learning against baselines including SFT and Model Merging. Curriculum Learning achieves the strongest results, showing that controlling how tasks are scheduled is crucial for effective multi-task learning.

Reinforcement learning (RL) has become an effective approach for improving large language models (LLMs) (Hurst et al., 2024; Liu et al., 2024; Dubey et al., 2024; Yang et al., 2024), particularly in structured domains such as math and coding where verifiable, rule-based rewards are available (Guo et al., 2025; Luo et al., 2025; Kimi-Team et al., 2025). Methods such as Group Relative Policy Optimization (GRPO) (Shao et al., 2024) show that even coarse learning signals can steer LLMs toward structured, chain-of-thought reasoning. However, most RL methods remain tailored to deterministically verifiable tasks, limiting their utility in open-ended domains such as general QA and creative writing. Moreover, training LLMs across multiple tasks remains challenging because it requires optimizing for diverse forms of feedback signals, including binary correctness checks in structured tasks and subjective, preference-based judgments in generative ones.

---

*Equal contribution. ♠Corresponding to: {jiaming.zhou, yingxue.zhang}@huawei.com

We address this challenge with OMNI-THINKER, a unified RL framework that enables LLMs to learn from both rule-based and generative supervision under a single policy. Building on *Reinforcement Learning with Verified Reward (RLVR)*, our method integrates symbolic verifiers with *LLM-as-a-Judge* evaluations (Zheng et al., 2023; Zhang et al., 2025) to handle subjective tasks. Our curriculum is forgetting-aware; it is guided by backward transfer (BWT), where BWT denotes test-performance backward transfer computed on a normalized, task-specific test metric. Ordering task training according to this signal yields effective curricula across heterogeneous domains. We show that the final accuracy of model after curriculum learning is well predicted by *forgettability* ranking, even under simplifying assumptions. Empirically, we observe complementary entropy dynamics, fine-tuning on creative writing tends to increase the model's output entropy, whereas training on verifier-supervised, structured tasks tends to decrease it; this trend is consistent with our BWT-guided choice to train structured tasks before open-ended ones. Across four domains, OMNI-THINKER improves generalization while reducing forgetting, with average gains of 6.2% over joint multi-task training and 12.4% over model merging, respectively.

Our key contributions are threefold. (1) We present OMNI-THINKER, a unified framework that trains a single policy across four diverse domains, using hybrid verifiable and preference-based rewards. (2) We develop a forgetting-aware curriculum based on backward transfer (BWT) linear ordering maximization over task-specific test performance to reduce forgetting, outperforming joint multi-task training and model merging. (3) We empirically analyze training dynamics through the lens of entropy, revealing that structured domains (math, coding) systematically decrease output entropy while open-ended domains (creative writing) increase it, thereby providing an explanatory link between entropy evolution and the effectiveness of BWT-guided curricula.

## 2 FRAMEWORK OVERVIEW

We introduce OMNI-THINKER as a unified reinforcement learning framework for large language models that integrates hybrid rewards with task scheduling guided by backward transfer. Unlike prior approaches that separate reasoning and generative domains, OMNI-THINKER maintains a single policy across heterogeneous tasks, including Math, Coding, General QA, and Creative Writing, while dynamically ordering training to minimize forgetting. The framework is instantiated using Multi-Task GRPO, augmented with both symbolic verifiers and LLM-as-a-Judge supervision, and a curriculum determined by accuracy- and entropy-based backward transfers.

### 2.1 NOTATION AND TRAINING OBJECTIVE

We give ourselves a vocabulary $\mathcal{V}$ with a special end-of-sequence token eos. The set of finite sequences of tokens is denoted $\mathcal{V}^*$; for any sequence $o \in \mathcal{V}^*$, its length is denoted $|o|$ and we say that $o$ is *complete* if $o_{|o|} = $ eos. A model, parameterized by $\theta$, defines a conditional distribution $\pi_\theta(y_t \mid y_{<t})$ for any given sequence of tokens $(y_t)_{t\in\mathbb{N}}$. It induces a policy $\pi_\theta^\otimes$ on token sequences defined by $\pi_\theta^\otimes(o \mid q, o_{<t_0}) := \prod_{t=t_0}^{|o|} \pi_\theta(o_t \mid q, o_{<t})$. We adopt a multi-task RL (MTRL) formulation: a task is a couple $T = (\mathcal{D}, R)$ where $\mathcal{D}$ is a dataset of prompts and $R(q, o)$ is a task-specific reward function. Given a set of $K$ tasks $\mathcal{T} = \{T_1, \ldots, T_K\}$, the goal is to learn a unified policy $\pi_\theta$ that maximizes the expected reward over the task distribution:

$$\max_\theta \; \mathcal{J}(\theta) = \mathbb{E}_{(\mathcal{D},R)\sim P(\mathcal{T})} \left[ \mathbb{E}_{q\sim\mathcal{D}, o\sim\pi_\theta^\otimes(\cdot|q)} [R(o)] \right], \tag{1}$$

where $P(\mathcal{T})$ is a task sampling distribution, which determines task exposure during training.

In order to train $\pi_\theta$ to maximize the objective $\mathcal{J}$, we extend the GRPO (Guo et al., 2025) algorithm to the multi-task setting by jointly optimizing over task-specific reward signals and reference policies. For each input prompt $q$, GRPO samples a group of outputs $\{o_{q,1}, o_{q,2}, \cdots, o_{q,G}\}$ from the old policy $\pi_{\theta_{old}}$. A task-specific reward function $R_k(q, o)$ scores each output. The policy $\pi_\theta$ is updated to maximize expected return while controlling divergence from a reference policy.

We define the policy ratio $\rho_{q,i,t}$ and the normalized advantage estimate $\hat{A}_{q,i,t}$ as follows:

$$\mu_q = \text{mean}\big(\{R_k(q, o_{q,i})\}_{i=1}^G\big), \qquad \sigma_q = \text{std}\big(\{R_k(q, o_{q,i})\}_{i=1}^G\big), \tag{2}$$

$$\rho_{q,i,t} = \frac{\pi_\theta(o_{q,i,t} \mid q, o_{q,i,<t})}{\pi_{\theta_{old}}(o_{q,i,t} \mid q, o_{q,i,<t})}, \qquad \hat{A}_{q,i,t} = \frac{R_k(q, o_{q,i}) - \mu_q}{\sigma_q}. \tag{3}$$

This allows us to write the MT-GRPO objective as

$$\mathcal{J}_{\text{MT-GRPO}}(\theta) = \mathbb{E}_{k \sim K, q \sim \mathcal{D}_k, \{o_{q,i}\}_{i=1}^{G} \sim \pi_{\theta_{\text{old}}}^{\otimes}(\cdot | q_k)}$$

$$\frac{1}{G} \sum_{i=1}^{G} \frac{1}{|o_{q,i}|} \sum_{t=1}^{|o_{q,i}|} \left\{ \min \left[ \rho_{q,i,t} \hat{A}_{q,i}, \text{clip}\left(\rho_{q,i,t}, 1-\epsilon, 1+\epsilon\right) \hat{A}_{q,i} \right] - \beta_k \mathbb{D}_{KL}\left[\pi_\theta || \pi_{ref}\right] \right\}, \tag{4}$$

where

$$\mathbb{D}_{KL}\left[\pi_\theta || \pi_{ref}\right] = \frac{\pi_{ref}(o_{q,i,t}|q, o_{q,i,<t})}{\pi_\theta(o_{q,i,t}|q, o_{q,i,<t})} - \log \frac{\pi_{ref}(o_{q,i,t}|q, o_{q,i,<t})}{\pi_\theta(o_{q,i,t}|q, o_{q,i,<t})} - 1. \tag{5}$$

The clipping parameter $\epsilon$ stabilizes updates by keeping policy ratios within a bounded range, following the PPO approach (Schulman et al., 2017). The KL divergence term regularizes the new policy towards the reference policy $\pi_{\text{ref}}$, weighted by a task-specific coefficient $\beta_k$.

## 2.2 HYBRID REWARDS

We design a hybrid reward system that unifies reinforcement learning across both structured reasoning tasks and open-ended generative domains.

**Verifiable Supervision.** For tasks with objective correctness signals, such as symbolic math and code generation, we define binary rewards based on symbolic matches, test case results, or other deterministic evaluators depending on the tasks.

**Short-Form Open-Ended Supervision.** For language tasks with known or extractable ground-truth answers such as general question answering (QA), we reformulate queries into open-ended prompts and incorporate distractor responses (LLM-generated plausible but incorrect answers) into the context. Instead of labeling options, we prompt the model to reason using the `<think>`...`</think>` format and to output answers within `<answer>`...`</answer>` tags. Responses are evaluated with a binary reward based on string matching or set membership against reference answers, thereby encouraging semantic grounding and mitigating shallow pattern memorization. We find that conditioning the LLM on a diverse set of candidate options, including one correct answer and multiple distractors, is key to steadily improving general-domain reasoning while reducing susceptibility to random guessing or reward hacking, compared to directly prompting the model to generate open-ended answers during training without the augmented context.

**Long-Form Open-Ended Supervision.** For subjective tasks lacking ground truth (e.g., dialogue, writing), we use an *LLM-as-a-Judge* (Chen et al., 2025) to assign a scalar reward based on rubric-aligned pairwise preferences between candidate outputs. This enables learning in domains where symbolic correctness is insufficient or intractable. This prompt-based approach leverages recent advances in the general reasoning capabilities of LLMs, using generated chain-of-thoughts to elicit a ternary reward signal, preferred, tie, or dispreferred, without requiring large-scale preference data collection and reward model training.

Together, these components form a unified hybrid reward scheme: verifiable rewards ensure correctness where possible and generative-based signals cover subjective domains. This design enables reinforcement learning to scale across diverse tasks, from reasoning to open-ended generation.

## 2.3 JOINT TRAINING AND CURRICULUM LEARNING

In practice, a maximization step of the training objective $\mathcal{J}_{\text{MT-GRPO}}$ requires a batch $B$ of prompts sampled from $\bigcup_{k=1}^{K} \mathcal{D}_k$ then sampling a batch of outputs $\{o_{q,i}\}_{i=1}^{G}$ for each $q \in B$. A multi-task schedule is defined as a sequence of batches $(B_s)_{s=1}^{s_{\max}}$ such that $\forall s \neq s', B_s \cap B_{s'} = \emptyset$ and $\bigcup_{s=1}^{s_{\max}} B_s = \bigcup_{k=1}^{K} \mathcal{D}_k$.

Two special cases are considered: *Joint Training* and *Curriculum Learning*. *Joint Training* consists in sampling each batch $B$ uniformly at random among all samples (without replacement), disregarding their corresponding tasks: $\forall s, B_s \sim \mathcal{U}\left(\bigcup_{k=1}^{K} \mathcal{D}_k \setminus \bigcup_{s'<s} B_{s'}\right)$. *Curriculum Learning* on the other

hand consists of pure batches chosen from the same task until exhaustion of the task dataset. By pure, we mean that each batch is derived from only one task dataset: $\forall s, B_s \sim \mathcal{U}\left(\mathcal{D}_{k_s} \setminus \bigcup_{s' < s} B_{s'}\right)$ for some task schedule $(k_s)_{s \in \{1, \cdots, s_{\max}\}}$. A Curriculum is described by a permutation $\sigma \in \mathfrak{S}_K$ of the tasks, with $\mathfrak{S}_K$ the set of permutation of $\{1, \cdots, K\}$.

## 3 BACKWARD TRANSFER FOR TASK-SCHEDULING

We intend to use Backward Transfers (BWT) to guide our choice of curriculum. Following Lopez-Paz & Ranzato (2017) it is defined as follows.

**Definition 1** (Backward Transfer Matrix). *Let $\theta_0$ be a set of initial parameters of a model $\pi_\theta$ and let $\theta(\theta_0, T)$ be the set of parameters obtained after training $\pi_\theta$ on task $T$ starting from $\theta_0$. Write $\mathrm{Acc}(\theta, T)$ the accuracy of model $\pi_\theta$ on task $T$. The backward transfer matrix is defined by*

$$\mathrm{BWT}_{ij}(\theta_0) := \log \mathrm{Acc}(\theta(\theta_0, T_j), T_i) - \log \mathrm{Acc}(\theta_0, T_i). \tag{6}$$

### 3.1 A PRIORI PREDICTION OF TERMINAL ACCURACIES UNDER CONSTANT BWT

Our goal is to *choose a curriculum order $\sigma$ a priori* by predicting the terminal per–task accuracies *without* training all permutations. We propose a simple predictive model in which (i) inter–task backward transfers are treated as constant in log–accuracy, and (ii) training on the full dataset of a task saturates its self–accuracy. Under these assumptions, terminal accuracies for any order $\sigma$ become computable from quantities measured once at initialization.

**Setup.** Using notations from Section 2.3, let $\theta_0$ denote the parameters of the pre–trained model $\pi_\theta$, and let $\theta_s$ be the parameters after $s$ optimization steps following a curriculum order $\sigma \in \mathfrak{S}_K$.

---

**Algorithm 1:** Final Accuracy under Assumptions 1 and 2

**Input:** $\mathrm{BWT} \in \mathbb{R}^{K \times K}$.
$\quad\quad\ a_{\mathrm{init}} := (\mathrm{Acc}(\theta_0, T_k))_{k=1}^K \in \mathbb{R}^K$.
$\quad\quad\ $ Curriculum $\sigma \in \mathfrak{S}_K$

$a \leftarrow a_{\mathrm{init}}$;
**for** $j = 1$ **to** $K$ **do**
$\quad$ $k \leftarrow \sigma(j)$;
$\quad$ $a_k \leftarrow a_{\mathrm{init},k}$;
$\quad$ **for** $i = 1$ **to** $K$ **do**
$\quad\quad$ $a_i \leftarrow a_i \times \exp(\mathrm{BWT}_{ik})$;
$\quad$ **end**
**end**
**return** $a$;

---

**Assumption 1** (Constant off–diagonal BWT in log–accuracy). *For all $i \neq j$ and all optimization steps $s$ along the schedule,*

$$\mathrm{BWT}_{ij}(\theta_s) = \mathrm{BWT}_{ij}(\theta_0). \tag{7}$$

**Assumption 2** (Task–wise saturation). *Let $\{B_s\}$ be the sequence of mini–batches processed along the schedule. If, between steps $s_1$ and $s_2$, the full dataset $\mathcal{D}_k$ of task $T_k$ has been seen, then accuracy saturates on task $T_k$ to the same accuracy as training from $\theta_0$:*

$$\bigcup_{s=s_1}^{s_2} B_s = \mathcal{D}_k \quad \Rightarrow \quad \mathrm{Acc}(\theta_{s_2}, T_k) = \mathrm{Acc}\big(\theta(\theta_0, T_k), T_k\big). \tag{8}$$

**Theorem 1.** *Under Assumptions 1 and 2, for any curriculum order $\sigma \in \mathfrak{S}_K$ starting from $\theta_0$, the terminal accuracies $\{\mathrm{Acc}(\theta_{s_{\max}}, T_k)\}_{k=1}^K$ given the initialization accuracies $\{\mathrm{Acc}(\theta_0, T_k)\}_{k=1}^K$, the $\mathrm{BWT}(\theta_0)$ and a curriculum $\sigma$ is exactly the output of Algorithm 1*[1]

**Reasonableness and limitations.** Assumption 1 deliberately abstracts away major contributors to inter-task transfer—such as domain overlap, optimization stochasticity, and the entropy dynamics discussed in Sections 5.2–5.3. It is therefore intentionally a *simplified* and *a priori* approximation used solely to obtain coarse curriculum predictions without training all task permutations. Assumption 2 ensures that each task's self-accuracy resets to its saturated value once its dataset is fully processed; this reflects our empirical setting, where datasets are sufficiently large and optimization remains well-conditioned.

---

[1]See appendix A.6 for a proof.

Crucially, these assumptions do *not* imply that the cumulative effect of sequential training is order-invariant. As formalized in Theorem 1, accuracies of earlier tasks continue to accumulate multiplicative backward-transfer effects from all *later* tasks, making the final outcome explicitly order-dependent unless all off-diagonal BWT terms vanish. The assumptions therefore serve as a lightweight but tractable model capturing the *coarse* structure of curriculum effects, while remaining simple enough to evaluate without exhaustive training. Predictions may deviate in magnitude—including occasional values exceeding 1 in accuracy due to log-space accumulation—but remain informative for ranking curricula and are empirically refined by entropy-based analysis.

## 3.2 CURRICULUM CHOICE VIA LINEAR ORDERING MAXIMIZATION

Algorithm 1 admits the following closed form for the predicted terminal accuracies:

$$\log \mathrm{Acc}(\sigma) - \log \mathrm{Acc}(\mathrm{Id}) = \sum_{i<j} \left( \Phi_\sigma^{-1} \mathrm{BWT}\, \Phi_\sigma \right)_{ij}, \quad \text{with} \quad \Phi_{\sigma,ij} = \mathbf{1}_{i=\sigma(j)}. \tag{9}$$

In words, curriculum reordering acts by permuting the BWT matrix with $\Phi_\sigma$, and the gain relative to the identity schedule is simply the sum of the upper–triangular entries of the permuted matrix.

Given an aggregated score of the form

$$\mathcal{S} := \sum_{T \in \mathcal{T}} \alpha_T \log \mathrm{Acc}(\theta, T), \tag{10}$$

---

**Algorithm 2:** Greedy BWT-LOM Curriculum

**Input:** BWT matrix $\mathrm{BWT} \in \mathbb{R}^{K \times K}$
$\sigma \leftarrow$ empty list;
**while** *there are unvisited tasks* **do**
  $k^* \leftarrow \arg\max_{k \notin \sigma} \sum_{i \notin \sigma \cup \{k\}} \mathrm{BWT}_{ik}$;
  append $k^*$ to $\sigma$;
**end**
**return** $\sigma$;

---

identifying the best task order amounts to solving a *Linear Ordering Problem* (LOP). see (Floudas & Pardalos, 2008) for an overview. This problem is known to be NP–hard, but for small numbers of tasks ($K$) it can be solved exactly. For larger $K$, a wide range of approximation algorithms and heuristics exist. A simple heuristic is to rank tasks by a *forgettability score*: $F_k := \alpha_k \sum_{i \neq k} \mathrm{BWT}_{ik}$. Intuitively, ordering tasks by decreasing $F_k$ prioritizes those that exert the least destructive interference on others (or even provide positive transfer), thereby reducing overall forgetting. Our curriculum orders tasks by decreasing column mean of BWT.

## 3.3 COMPUTATIONAL OVERHEAD OF CURRICULUM SCHEDULING

Constructing the curriculum requires estimating a BWT matrix, which involves one RL fine-tuning run per task plus forward-pass evaluations over all ordered task pairs. The dominant cost is the $K$ single-task trainings: each has the same order of complexity as one curriculum pass, since all curricula process every task dataset to saturation. Thus, the total overhead is approximately equivalent to one additional curriculum run.

Evaluation is comparatively negligible. Estimating all $K^2$ BWT entries to accuracy $\epsilon$ scales as $O(K^2/\epsilon^2)$ forward passes, which are substantially cheaper than RL updates. A limitation arises only when artificially increasing task granularity: if a fixed dataset is partitioned into many subtasks, the $K^2$ evaluation term may become non-negligible. Overall, the proposed method yields a near-optimal curriculum estimate at the cost of roughly two curriculum executions, offering a principled trade-off between computational efficiency and scheduling quality.

## 4 EXPERIMENTAL SETUP

**Training Datasets.** We curate a multi-domain training dataset covering Math, Coding, General QA, and Creative Writing, with each domain selected to support hybrid reward functions and robust evaluation. For **Math**, we begin with the OpenR1-Math (HuggingFace, 2025) dataset, retaining only word problems and excluding questions that require visual reasoning. We further subsample 12,000 examples to fit our compute budget. For **Coding**, data is sourced from the code-r1-12k (Liu & Zhang, 2025) dataset, with outliers exceeding 1024 tokens removed. Each entry includes a code prompt and JSON-formatted unit tests for automatic validation. For **General QA**, we subsample 5,500 queries from from SuperGPQA (Du et al., 2025) dataset, proportionally by question category.

Each sample comprises a factual question paired with a plain-text answer. We then generate 15 additional confusion options while making sure the uniqueness of correctness by prompting an LLM. The **Creative Writing.** domain leverages 6,650 conversations from Nitral AI's ShareGPT dataset (Nitral-AI, 2024), focused on single-turn completions. Samples exceeding two dialogue turns are filtered out, and responses are judged via an *LLM-as-a-Judge* framework.

**Evaluation.** We assess performance in each domain using dedicated benchmarks aligned with the task's evaluation criteria. **Math**: accuracy on AIME24 (MAA, 2024), AMC23 (MAA, 2023), Gaokao2023EN (Liao et al., 2024), MATH-500 (Hendrycks et al., 2021), MinervaMath (Lewkowycz et al., 2022), and OlympiadBench (He et al., 2024). **Coding**: pass@1 on BigCodeBench (Complete-Full) (Zhuo et al., 2024) and LiveCodeBench (24Oct–25Jan) (Jain et al., 2024). **General QA**: exact-match accuracy on MMLU-Pro (Wang et al., 2024). **Creative Writing**: win rate on the *role-play* and *creative writing* subsets of MT-Bench (Zheng et al., 2023), against GPT-4 (*pre-gen, June 16, 2023*).

**Baselines.** We use Qwen2.5-7B-Instruct (Yang et al., 2024) as the base model for all experiments, owing to its strong instruction-following capability, which makes it well-suited for reinforcement learning on both structured reasoning and open-domain QA tasks. **Supervised Fine-Tuning (SFT):** In order to have a meaningful comparison with GRPO, we adopt a similar self-sampled data curation and fine-tuning approach with Rejection sampling Fine-Tuning (Yuan et al., 2023). We first prompt the base model to generate 128 chain-of-thought responses for our training dataset to ensure we end up with at least one correct response for most queries, then filter them based on the same accuracy reward signals used in GRPO training. We then perform sft on base model using these self-distilled responses. This provides a strong on-policy learning baseline that incorporates explicit reasoning steps through self-distillation from the base model. **Model Merging:** We employ *TIES-Merging* (Wu et al., 2025) as our model-merging baseline. It is a simple yet effective method designed specifically for the multi-task setting that takes into consideration the interference between parameters from models trained on individual tasks during the merging process. It has demonstrated superior performance in multi-task learning compared to linear and task arithmetic approaches (Yadav et al., 2023). To begin with, we conduct single-task GRPO training using individual task datasets and collect the model weights of the best checkpoints with the help of a validation set for each training run. We then merge the four single-task models using a scaling value $\lambda = 1$.

## 5 RESULTS AND DISCUSSION

### 5.1 MAIN RESULTS: SCALING MULTI-TASK LLM POST-TRAINING WITH OMNI-THINK

We evaluate OMNI-THINKER across four diverse domains: Coding, Math, General QA, and Creative Writing, to assess how reinforcement learning with rule-based verifiable rewards and generative supervision supports multi-task generalization. BWT matrix is computed following equation 6, then Algorithm 1 is used to predict the accuracy of the model after curriculum learning, Appendix A.2.2 for details. The predicted best curriculum using Algorithm 2 is Code → Math → QA → Writing while the worst is Writing → QA → Math → Coding.

Figure 1 shows that Curriculum Multi-Task Learning with GRPO consistently yields the best results. Table 1 further details how these gains vary by benchmarks.

In **Math**, Curriculum Learning (CL) achieves the highest average performance at 59.6%, with the clearest gains on more complex reasoning tasks such as MinervaMath and OlympiadBench. These benchmarks benefit from strong rule-based reward signals and backward-transfer-guided task ordering. In contrast, datasets like AMC23 show minimal change because their relatively high baseline scores likely reflect smaller question sets and potential pretraining overlap rather than robust multi-step problem-solving.

In **General QA**, CL again performs best (52.2%), followed by Model Merging (49.8%) and Mixed Training GRPO (48.8%). These improvements are driven by our Short-Form Open-Ended Supervision strategy: instead of generating responses in a fully open-ended and unconstrained fashion, the model is trained to produce complete answer strings given a diverse set of candidate responses, enabling

Table 1: Performance across benchmarks. **ST** = Single-Task RL (e.g., **ST Math** = RL trained only on math). **MM** = Model Merging. **JT** = Joint Training. **CL** = Curriculum Learning. Domains include Math (7 sets), MMLU-Pro (9 categories), Coding (2 sets), and Creative Writing (MT-Bench).Highlighted are the top first, second, and third results.

| Eval Task | Base Model | ST Coding | ST Math | ST QA | ST Writing | SFT | MM | JT | CL_best |
|---|---|---|---|---|---|---|---|---|---|
| **Math** | | | | | | | | | |
| AIME24 | 18.0 | 13.3 | 14.7 | 14.0 | 15.3 | 16.7 | 10.0 | 11.3 | 15.3 |
| AMC23 | 57.5 | 57.5 | 60.0 | 62.5 | 61.0 | 62.0 | 56.0 | 51.0 | 70.0 |
| Gaokao2023en | 73.0 | 74.3 | 76.1 | 74.0 | 75.6 | 74.3 | 74.8 | 76.6 | 77.1 |
| MATH500 | 78.2 | 78.8 | 80.4 | 75.4 | 79.2 | 76.8 | 79.8 | 77.6 | 81.0 |
| MinervaMath | 64.3 | 64.0 | 66.5 | 63.2 | 61.8 | 65.1 | 66.2 | 68.4 | 71.7 |
| OlympiadBench | 42.1 | 43.0 | 43.7 | 41.3 | 43.0 | 43.0 | 41.8 | 43.6 | 47.4 |
| **Average** | 55.5 | 55.1 | 56.9 | 55.1 | 56.0 | 56.3 | 54.8 | 54.8 | 60.4 |
| **General QA** | | | | | | | | | |
| Biology | 57.6 | 56.8 | 52.3 | 67.4 | 59.0 | 66.3 | 65.6 | 67.2 | 68.8 |
| Business | 33.5 | 39.0 | 25.6 | 58.7 | 33.0 | 48.2 | 59.8 | 49.8 | 47.5 |
| Chemistry | 35.8 | 31.8 | 27.3 | 47.7 | 38.3 | 44.1 | 42.5 | 42.1 | 50.7 |
| CS | 53.7 | 48.1 | 50.2 | 55.1 | 52.0 | 53.7 | 53.9 | 58.8 | 59.3 |
| Economics | 42.7 | 49.2 | 38.7 | 62.9 | 44.9 | 59.6 | 62.0 | 56.8 | 62.1 |
| Engineering | 28.3 | 31.3 | 20.4 | 37.5 | 26.6 | 37.8 | 38.1 | 35.8 | 37.1 |
| Health | 46.7 | 46.2 | 45.2 | 51.0 | 47.2 | 45.7 | 52.7 | 50.7 | 57.1 |
| History | 37.3 | 33.3 | 34.7 | 47.2 | 38.6 | 33.9 | 47.3 | 43.3 | 45.7 |
| Law | 23.2 | 24.0 | 20.6 | 27.9 | 23.3 | 26.8 | 26.6 | 27.5 | 29.7 |
| Math | 55.4 | 52.6 | 50.4 | 59.3 | 56.3 | 57.4 | 58.3 | 59.2 | 61.2 |
| Other | 44.3 | 40.0 | 39.7 | 51.0 | 43.9 | 46.4 | 51.8 | 49.9 | 53.3 |
| Philosophy | 36.9 | 34.3 | 33.3 | 43.9 | 35.5 | 38.2 | 41.5 | 42.1 | 42.9 |
| Physics | 41.1 | 37.4 | 30.8 | 53.7 | 41.6 | 49.8 | 46.7 | 48.0 | 55.6 |
| Psychology | 50.9 | 51.5 | 45.4 | 60.2 | 51.8 | 59.0 | 59.4 | 59.3 | 61.8 |
| **Average** | 41.5 | 40.1 | 37.9 | 51.3 | 42.0 | 47.8 | 49.8 | 48.8 | 52.2 |
| **Coding** | | | | | | | | | |
| BigCodeBench | 46.5 | 50.4 | 46.7 | 47.1 | 46.8 | 44.5 | 48.1 | 47.2 | 49.5 |
| LiveCodeBench | 12.7 | 21.8 | 13.1 | 13.8 | 13.3 | 14.2 | 20.8 | 17.0 | 21.3 |
| **Average** | 29.6 | 36.1 | 29.9 | 30.5 | 30.1 | 29.3 | 34.4 | 32.1 | 35.4 |
| **Creative Writing** | | | | | | | | | |
| MT-Bench (Writing) | 74.2 | 71.6 | 74.2 | 63.0 | 78.3 | 44.2 | 67.5 | 83.0 | 84.2 |

the effective application of verifiable reward through simple string matching when training general-domain tasks.

For **Code Generation**, CL achieves 35.4%, slightly ahead of Model Merging. Notably, we only evaluate on the subset of LiveCodeBench(24Oct-25Jan) problems released after Qwen2.5's data cutoff, which ensures that these are unseen test items. This setup highlights CL's significant generalization gains on novel problems, explaining the larger improvements on LiveCodeBench relative to static benchmarks like BigCodeBench, where data overlap is more likely.

In **Creative Writing**, the introduction of our Long-Form Open-Ended Supervision strategy, employing the *LLM-as-a-Judge* framework for pairwise evaluation, results in significant performance boosts (Curriculum-Guided at 84.2% and Joint MT at 83.00%), underscoring the advantage of our generative reward approach in subjective, open-ended tasks.

These results support our central hypothesis: The OMNI-THINKER Framework, BWT-guided Curriculum Learning with hybrid rewards, enables a single unified policy to scale across structured and open-ended tasks alike, without relying on interleaving *RLVR* on reasoning tasks and fine-tuning non-reasoning tasks.

## 5.2 ENTROPY DYNAMICS: DISCUSSION

Comparison between accuracies predictions to the actual test evaluation results for various curricula is depicted on Table 3. Predicted accuracies using test set backward transfers are surprisingly precise considering our simplifying assumptions, especially for the top curriculum. We now discuss an identified cause of discrepancies.

We define the token-wise entropy of a policy $\pi_\theta$ on task $T_k$ ti measures the average per-token uncertainty of the policy across task samples.

$$\mathcal{E}(\theta, T_k) := -\mathbb{E}_{q\sim\mathcal{D}_k, o\sim\pi_\theta^\otimes(\cdot|q)} \frac{1}{|o|} \sum_{t=1}^{|o|} \sum_{v\in\mathcal{V}} \pi_\theta(v \mid q, o_{<t}) \log \pi_\theta(v \mid q, o_{<t}). \tag{11}$$

Entropy has been shown to drop during GRPO fine-tuning on reasoning, coding and more generally verified rewards (Rastogi et al., 2025; Cui et al., 2025; Yu et al., 2025) with a correlation to higher accuracy until reaching a breaking point. Long training requires extra care towards entropy scaling either via regularization (Shen, 2025) or dynamic temperature scaling. Our multi-task setting differs in two key aspects compared to the above references.

Table 2: Test performance (%) of single-task RL fine-tuning on General QA and Creative Writing respectively under different generation temperature (T) in training.

| Eval Task | General QA | | Creative Writing | |
|---|---|---|---|---|
| | T=1.0 | T=1.2 | T=1.0 | T=0.1 |
| Math | 55.1 | 54.8 | 55.6 | 57.8 |
| Coding | 30.4 | 26.6 | 30.1 | 27.4 |
| General QA | **51.4** | **13.9** | 42.0 | 44.9 |
| Creative Writing | 63.0 | 66.7 | **78.3** | **82.5** |

First, we are using hybrid rewards with both verified and generative components. The Creative Writing task is generative and is expected to increase entropy. Indeed, Wang et al. (2025) account for so-called *forking* tokens corresponding to structural choices of the output capturing most of the entropy. Reasoning tasks tend to require highly causal token sequences hence few forking tokens (low entropy) while generative tasks may allow more logical cuts at inference (higher entropy).

Second, we train on multiple domains compared to mostly single-domain analysis in the references above. It is unclear a priori whether entropy decrease propagates from task to task. Agarwal et al. (2025) show that fine-tuning to reduce entropy suffices to improve performance on multiple domains. We hypothesize that models implicitly learn to emulate lower or higher temperatures as a mechanism to regulate entropy. Thus, even when two domains are sufficiently distinct for knowledge transfer to fail, the *entropy dynamics* may still be measurable across tasks. While several possible mechanism are conceivable, we do not observe evidence supporting any specific one.

Table 3: Comparison of empirical and predicted test accuracies (%). Each task column reports **Test** vs. **Predicted** accuracy for a given curriculum order. Standard deviations are rounded up.

| Curriculum | Math | | Coding | | QA | | Writing | |
|---|---|---|---|---|---|---|---|---|
| | Test | Pred | Test | Pred | Test | Pred | Test | Pred |
| $\mathcal{C}\,\mathcal{M}\,\mathcal{Q}\,\mathcal{W}$ | 60.4±0.3 | 57.3 | 35.4±0.3 | 35.4 | 52.2±0.1 | 51.9 | 84±3 | 78.4 |
| $\mathcal{Q}\,\mathcal{M}\,\mathcal{W}\,\mathcal{C}$ | 59.3±0.3 | 55.6 | 31.6±0.3 | 36.7 | 39.0±0.1 | 38.8 | 79±3 | 78.4 |
| $\mathcal{Q}\,\mathcal{W}\,\mathcal{C}\,\mathcal{M}$ | 60.4±0.3 | 57.6 | 31.9±0.3 | 33.9 | 36.3±0.1 | 38.8 | 82±3 | 73.1 |
| $\mathcal{W}\,\mathcal{Q}\,\mathcal{M}\,\mathcal{C}$ | 56.6±0.3 | 55.5 | 32.7±0.3 | 36.1 | 22.6±0.1 | 38.4 | 75±3 | 62.1 |

## 5.3 ENTROPY DYNAMICS: EMPIRICAL SUPPORT

The intuitions laid out in the previous section are empirically supported by two experiments.

We define the entropy change matrix as

$$H_{ij} := \frac{\mathcal{E}(\theta(\theta_0, T_i), T_j)}{\mathcal{E}(\theta_0, T_j)} - 1 \tag{12}$$

and compute it, see Figure 2. We observe that Math and Coding decrease entropy for all tasks (as previously observed for Verified Rewards) while Creative Writing increases entropy. Also, entropy change seems to depend primarily on the source task type, secondarily on the target task type, but not on their domain overlap.

We fine-tune the base model on QA and Writing task with different choices of temperature to emulate the effect of entropy modifications due to fine-tuning on entropy-increasing or entropy-decreasing tasks. On the one hand, Writing is thought to benefits from temperature lowering coming from other tasks, we thus train the base model on writing with a lower temperature and expect the model to close the gap compared to the best Curriculum-trained model when evaluated with zero temperature.

On the other hand, QA is thought to performs worse than expected in the worst curriculum due to the increased entropy coming from the trainig on Writing. We fine-tune the base model on QA with a higher temperature, then evaluate with zero temperature, and expect QA performance to drop toward the low performance of the worse curriculum learning. The results on table 2 shows that indeed the case.

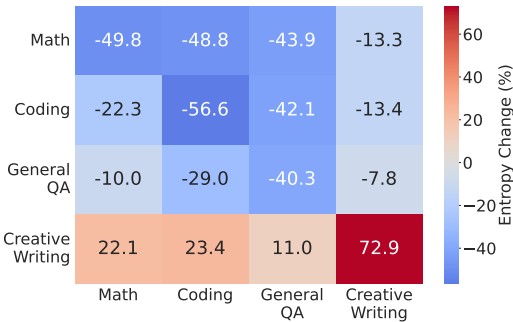

Figure 2: Validation Set Entropy Change Matrix. Each *column* corresponds to the task being evaluated, and each *row* corresponds to the task the model was fine-tuned on. Each cell reports the relative change in entropy on the evaluation task after training on the corresponding source task, revealing how entropy shifts arising from one task carry over to others.

## 6 RELATED WORK & LIMITATIONS

**Large Language Models and Multi-Task Learning**   Early work like (Sanh et al., 2021) showed that multi-task prompted training can encourage zero-shot generalization. Dong et al. (2023) further analyzed how mixing SFT data across domains can cause performance conflicts and forgetting, proposing Dual-stage Mixed Fine-tuning to alleviate these effects. However purely supervised objectives often encourage memorization rather than transferable reasoning. The Qwen3 model series (Yang et al., 2025) employs a four-stage post-training pipeline in the order of reasoning, non-reasoning, and general-domain under a mix of supervised fine-tuning and reinforcement learning. In comparison, the post-training process for Command-A (Cohere et al., 2025) alternates between training multiple expert models separately and merging the experts' parameters into a "Soup Model" during its SFT and RL steps, before the model undergoes a polishing phase of preference alignment. In contrast, our work integrates multi-task learning directly into a single RL framework. Its backward-transfer-guided curriculum orders tasks from least to most forgettable, drawing on continual learning insights (Lopez-Paz & Ranzato, 2017) to reduce interference and maintain stable cross-task performance.

**Large Language Models and Reinforcement Learning**   Reinforcement Learning with Verified Rewards has demonstrated effectiveness for tasks with deterministic correctness signals such as math or code generation (Lambert et al., 2024; Shao et al., 2024; Kimi-Team et al., 2025; Guo et al., 2025). Recent frameworks like General-Reasoner (Ma et al., 2025), Nemotron-Crossthink (Akter et al., 2025) and X-REASONER (Liu et al., 2025) expand this to broader reasoning by blending multi-domain corpora and structured answer templates. However, these tasks still largely remain largely confined to verifiable STEM problems or multiple-choice formats, leaving open-ended generation, such as creative writing, insufficiently addressed. To bridge this gap, Su et al. (2025) propose a generative reward model (GRM) to replace rule-based signals. Although this improves RL and makes it applicable to general-domain QA when references exist, the approach is still restricted to verifiable tasks. In contrast, our approach integrates hybrid verifiable and preference-based rewards within a single RL loop, enabling consistent optimization across both structured and open-ended tasks. Moreover, our curriculum design, guided by backward transfer, helps maintain stable cross-task performance even for tasks lacking deterministic evaluation criteria.

## 7 CONCLUSION

We presented OMNI-THINKER, a unified reinforcement learning framework that enables large language models to handle both structured and open-ended tasks under a single policy. By combining rule-based verifiable rewards and generative preference-based supervision, our method improves generalization while mitigating forgetting and interference. Our findings show that effective multi-task LLM post-training depends not only on reward design but also on how tasks are sequenced and optimized together. Ordering tasks from structured to open-ended domains based on backward transfer reduces forgetting and enhances cross-domain performance. Overall, OMNI-THINKER advances the goal of general-purpose LLMs that can learn from both verifiable and subjective feedback, bridging structured reasoning, open-ended question answering, and creative generation in a single post-training framework.

**Limitations:** Further work is needed to test this approach across a broader range of tasks and domains, including those that require logical reasoning over graph-structured data (Zhou et al., 2024) and knowledge-base retrieval (Dehghan et al., 2024), as well as more diverse open-ended domains such as mixed-initiative collaborative storytelling and co-creativity (Kreminski et al., 2024). Our discussion on entropy is consistent with the existing literature however we lack 1) experiments backing our intuition that forking tokens are clearly identifiable and more frequent in generative tasks is lacking 2) a direct measurement of temperature scaling due to fine-tuning. Furthermore, our discussion on entropy is mostly qualitative. A quantitative study of entropy effects could allow to avoid failure cases of our accuracy predictions hence of our task scheduler. Finally, we fully leverage BWT only for curriculum learning. A comprehensive framework combining BWT, entropy and gradient norm in a quantitative manner in the continuous limit would allow for dynamical joint training with mixed batches.

## ETHICS STATEMENT

We have carefully reviewed the ICLR Code of Ethics and affirm our adherence to it. Our work does not involve human subjects, sensitive data, or personally identifiable information. It does not raise concerns regarding privacy, security, discrimination, bias, or potential misuse. To the best of our knowledge, our study complies with ethical standards and does not present issues related to research integrity, conflicts of interest, or legal compliance.

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

## LLM USAGE STATEMENT

In accordance with the conference policy on large language model (LLM) usage, we acknowledge that LLMs were employed as general-purpose assistants during the preparation of this work. Specifically, we used LLMs for proof-reading, LaTeX formatting support, adaptation of short code or text snippets. We also used LLMs a search engine and to find weaknesses. The LLMs did not contribute to research ideation, experimental design, or substantive writing, and their role was limited to auxiliary assistance.

## A    APPENDIX

### A.1    REWARD ESTIMATION

Omni-Thinker employs a hybrid reward system combining rule-based correctness (math, code, QA) with preference-based supervision (creative writing) in a unified RL framework. We define task-specific reward functions as $R_k(q, o)$, where $o$ denotes the model output and $q$ is the prompt provided to the model. Each reward function captures domain-relevant correctness criteria, assessing whether $o$ satisfies symbolic constraints, passes execution tests, or is preferred over alternatives under subjective evaluation. While some rewards (e.g., math and code) are strictly deterministic, others, such as LLM-as-a-Judge comparisons, are inherently stochastic but executed at low decoding temperature to ensure stable and consistent supervision. All reward functions are designed to be domain-aware and automatable, supporting scalable reinforcement learning across both structured and generative tasks.

**Primary Rewards.**    Each task employs a tailored correctness criterion:

- **Math:** Elements of the math dataset are couples $(q, a_q)$ where $q$ is a prompt and $a_q$ is a token sequence stating the answer. We implement a $\texttt{verify}_{\texttt{math}}(o, a)$ function that combines regular expression and symbolic parser to checks that $a$ (or an equivalent acceptable answer) is in $o$ within tags $\texttt{<answer>}$. More formally

$$r_{\texttt{math}}(q, o) = \mathbb{1}\left\{\texttt{verify}_{\texttt{math}}(o, a_q) = \texttt{true}\right\}.$$

- **Code Generation:**    Each element of the code dataset is a tuple $(q, \texttt{unittest}_q, \texttt{test\_case}_q)$ where $q$ is a prompt, $\texttt{unittest}_q$ is a unit test function and $\texttt{test\_case}_q$ is a set of test cases. Given an output $o$ for prompt $q$, the generated code $o_{\texttt{ans}}$ is extracted from the output $o$ using regular expressions, the unit test $\texttt{unittest}_q(o_{\texttt{ans}}, x)$ is executed in a sandboxed environment for every test case $x \in \texttt{test\_case}_q$. More formally

$$r_{\texttt{code}}(q, o) = \prod_{x \in \texttt{test\_case}_q} \mathbb{1}\left\{\texttt{unittest}_q(o_{\texttt{ans}}, x)\right\}$$

- **General QA:** Each element of the dataset for General QA is a couple $(q, a_q)$ of prompt and answer. The reward is defined by extracting the answer $o_{\texttt{ans}}$ from the output $o$ using regular expressions and testing it against the ground truth $a$. More formally:

$$r_{\texttt{qa}}(q, o) = \mathbb{1}\left\{o_{\texttt{ans}} = a_q\right\}$$

which returns 1 if the predicted answer matches the ground-truth string exactly.

- **Creative Writing:** Each element of the dataset for Creative writing is a couple $(q, o_{\texttt{ref},q})$. Given an output $o$ on prompt $q$, the reward is computed by calling a fixed *LLM-as-a-Judge* model prompted to do a pairwise comparison between $o$ and $o_{\texttt{ref},q}$. More formally:

$$r_{\text{writing}}(q, o) = \begin{cases} 1.0 & \text{if } o \succ_q o_{\texttt{ref},q} \\ 0.5 & \text{if } o \sim_q o_{\texttt{ref},q} \\ 0.0 & \text{if } o \prec_q o_{\texttt{ref},q} \end{cases}$$

where $A \succ_q B$ means that the fixed *LLM-as-a-Judge* model prefered $A$ over $B$ for request $q$, and $A \sim_q B$ means it judges the answers as tied.

**Auxiliary Rewards.** To encourage structured outputs, we define formatting-based rewards shared across tasks:

$$r_{\texttt{format}}(q, o) = \mathbb{1}\{\texttt{tags\_valid}(o)\}$$

$$r_{\texttt{tags}}(q, o) = \frac{1}{4} \cdot |\texttt{tags\_present}(o)|$$

Here, `tags_valid` ensures proper nesting of `<think>` and `<answer>` tags, while `tags_present` counts required structural markers.

**Total Reward.** We define the total reward as a weighted sum over both primary and auxiliary reward components. Let $\mathcal{F}_k = \{r_k^{(1)}, r_k^{(2)}, \ldots, r_k^{(m)}\}$ denote the set of reward functions associated with task $k$, where each $r_k^{(j)}$ measures a different aspect of correctness. Given a model output $o$ and its associated evaluation context $\phi_k$, the total reward is computed as:

$$R_k(q, o) = \sum_{r \in \mathcal{F}_k} w_r \cdot r(q, o),$$

where $w_r \in [0, 1]$ denotes the task-specific weight for reward component $r$. If a component reward is undefined, e.g., due to malformed or unparsable output, it is omitted from the sum. Samples with no valid reward components are excluded from policy updates.

## A.2 RESULTS

### A.2.1 OUTPUT FORMAT MATTERS: FULL-TEXT ANSWERS ENHANCE GENERALIZATION

We examine how output format impacts generalization by comparing models trained to generate full-text answers versus selecting letter choices in multiple-choice QA (MCQ). Using GRPO, we train two single-task policies on the training set, one prompted to produce full-text final answers at the end of its chain-of-thought completions, and the other to output only letter choices (e.g., "A", "B", "C").

As shown in Figure 3, the model trained to output full-text answers achieves significantly better generalization when evaluated with free-form QA prompts on MMLU Pro (51% vs. 41%). While the letter-choice model slightly outperforms when evaluated strictly on MCQ prompts, the full-text model remains competitive across both prompt formats.

These results suggest that training with complete, semantically grounded answers encourages deeper reasoning, improving the model's ability to generalize beyond the specific format seen during training. In contrast, letter-choice training risks overfitting to shallow pattern matching, reducing transferability to realistic QA settings that often require articulated responses.

Figure 3: Models trained to generate full-text answers outperform those trained to select letter choices. This format promotes deeper semantic understanding over shallow pattern matching or guessing.

### A.2.2 CURRICULUM ACCURACIES PREDICTIONS

On Figure 4 are provided BWT matrix for validation sets and test sets. We compute ranking of curricula using both, see table 4. We observe that he Forgettability heuristic provides the second best prediction on test BWT, but exact LOM yields a potentially better candidate. Using validation BWT, the curriculum Code-Math-QA-Writing comes second but the proposed exact LOM should be rejected based on our entropy discussion.

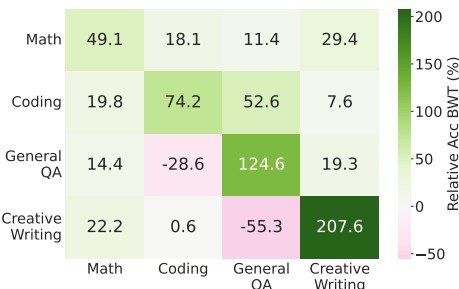 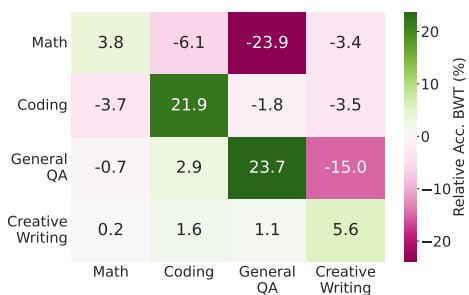

Figure 4: Accuracy backward transfer matrices on the validation set (left) and test set (right). Rows indicate the *source task* the model was fine-tuned on, and columns indicate the *evaluation task*. Each cell reports the relative accuracy change on the evaluation task after single-task fine-tuning on the corresponding source task, illustrating how training on one domain helps or harms performance on others.

Table 4: Curricula ranked by final geometrical average relative improvement

$$\overline{\Delta_{\mathrm{rel}}} := \frac{1}{K} \log \prod_{k=1}^{K} \frac{\mathrm{Acc}(\theta_{s_{\max}}, T_k)}{\mathrm{Acc}(\theta_0, T_k)} \tag{14}$$

on test set (left block) and eval set (right block), with predicted accuracies (%) for each task at each order, and their ranking scores.

| | Test set order | | | | | Validation set order | | | | |
| Order | M | C | Q | W | $\overline{\Delta_{\mathrm{rel}}}$ | Order | M | C | Q | W | $\overline{\Delta_{\mathrm{rel}}}$ |
|---|---|---|---|---|---|---|---|---|---|---|---|
| $\mathcal{M\,C\,Q\,W}$ | 55.2 | 37.7 | 51.9 | 78.4 | 12.85 | $\mathcal{C\,W\,M\,Q}$ | 34.9 | 42.5 | 26.8 | 35.4 | 26.00 |
| $\mathcal{C\,M\,Q\,W}$ | 57.3 | 35.4 | 51.9 | 78.4 | 12.22 | $\mathcal{C\,M\,Q\,W}$ | 36.7 | 42.5 | 27.5 | 32.6 | 25.85 |
| $\mathcal{M\,Q\,C\,W}$ | 55.2 | 36.7 | 51.0 | 78.4 | 11.70 | $\mathcal{C\,Q\,W\,M}$ | 33.1 | 42.5 | 28.1 | 35.3 | 25.81 |
| $\mathcal{M\,Q\,W\,C}$ | 55.2 | 36.1 | 51.0 | 75.7 | 10.43 | $\mathcal{W\,C\,M\,Q}$ | 34.9 | 44.7 | 26.8 | 33.4 | 25.77 |
| $\mathcal{M\,C\,W\,Q}$ | 55.2 | 37.7 | 51.3 | 66.6 | 8.49 | $\mathcal{C\,M\,W\,Q}$ | 36.7 | 42.5 | 26.8 | 32.8 | 25.29 |
| $\mathcal{C\,M\,W\,Q}$ | 57.3 | 35.4 | 51.3 | 66.6 | 7.86 | $\mathcal{C\,W\,Q\,M}$ | 33.1 | 42.5 | 27.4 | 35.4 | 25.25 |
| $\mathcal{M\,W\,C\,Q}$ | 55.2 | 37.1 | 51.3 | 64.3 | 7.22 | $\mathcal{C\,Q\,M\,W}$ | 34.8 | 42.5 | 28.1 | 32.6 | 25.09 |
| $\mathcal{C\,W\,M\,Q}$ | 57.2 | 35.4 | 51.3 | 64.3 | 6.96 | $\mathcal{W\,C\,Q\,M}$ | 33.1 | 44.7 | 27.4 | 33.4 | 25.02 |
| $\mathcal{W\,M\,C\,Q}$ | 55.1 | 37.1 | 51.3 | 62.1 | 6.32 | $\mathcal{M\,C\,Q\,W}$ | 37.0 | 40.2 | 27.5 | 32.6 | 24.71 |
| $\mathcal{M\,W\,Q\,C}$ | 55.2 | 36.1 | 50.4 | 64.3 | 6.07 | $\mathcal{W\,M\,C\,Q}$ | 35.3 | 42.3 | 26.8 | 33.4 | 24.64 |
| $\mathcal{W\,C\,M\,Q}$ | 57.2 | 34.8 | 51.3 | 62.1 | 5.69 | $\mathcal{M\,C\,W\,Q}$ | 37.0 | 40.2 | 26.8 | 32.8 | 24.16 |
| $\mathcal{C\,Q\,M\,W}$ | 57.7 | 35.4 | 39.5 | 78.4 | 5.56 | $\mathcal{M\,W\,C\,Q}$ | 37.0 | 42.3 | 26.8 | 30.9 | 23.93 |
| $\mathcal{W\,M\,Q\,C}$ | 55.1 | 36.1 | 50.4 | 62.1 | 5.18 | $\mathcal{Q\,C\,W\,M}$ | 33.1 | 41.3 | 26.1 | 35.3 | 23.23 |
| $\mathcal{Q\,M\,C\,W}$ | 55.6 | 36.7 | 38.8 | 78.4 | 5.05 | $\mathcal{Q\,W\,C\,M}$ | 33.1 | 43.4 | 26.1 | 33.3 | 23.00 |
| $\mathcal{C\,Q\,W\,M}$ | 57.6 | 35.4 | 39.5 | 75.7 | 4.67 | $\mathcal{Q\,C\,M\,W}$ | 34.8 | 41.3 | 26.1 | 32.6 | 22.52 |
| $\mathcal{Q\,C\,M\,W}$ | 57.7 | 34.4 | 38.8 | 78.4 | 4.41 | $\mathcal{W\,Q\,C\,M}$ | 33.1 | 43.4 | 25.4 | 33.4 | 22.45 |
| $\mathcal{Q\,M\,W\,C}$ | 55.6 | 36.1 | 38.8 | 75.7 | 3.77 | $\mathcal{M\,Q\,C\,W}$ | 37.0 | 39.1 | 25.5 | 32.6 | 22.14 |
| $\mathcal{Q\,C\,W\,M}$ | 57.6 | 34.4 | 38.8 | 75.7 | 3.52 | $\mathcal{W\,M\,Q\,C}$ | 35.3 | 41.1 | 24.9 | 33.4 | 22.07 |
| $\mathcal{Q\,W\,M\,C}$ | 55.5 | 36.1 | 38.8 | 73.1 | 2.88 | $\mathcal{M\,Q\,W\,C}$ | 37.0 | 41.1 | 25.5 | 30.8 | 21.91 |
| $\mathcal{Q\,W\,C\,M}$ | 57.6 | 33.9 | 38.8 | 73.1 | 2.25 | $\mathcal{Q\,W\,M\,C}$ | 33.5 | 41.1 | 26.1 | 33.3 | 21.87 |
| $\mathcal{C\,W\,Q\,M}$ | 57.6 | 35.4 | 39.0 | 64.3 | 0.31 | $\mathcal{Q\,M\,C\,W}$ | 35.2 | 39.1 | 26.1 | 32.6 | 21.39 |
| $\mathcal{W\,C\,Q\,M}$ | 57.6 | 34.8 | 39.0 | 62.1 | -0.97 | $\mathcal{M\,W\,Q\,C}$ | 37.0 | 41.1 | 24.9 | 30.9 | 21.35 |
| $\mathcal{W\,Q\,M\,C}$ | 55.5 | 36.1 | 38.4 | 62.1 | -1.48 | $\mathcal{W\,Q\,M\,C}$ | 33.5 | 41.1 | 25.4 | 33.4 | 21.31 |
| $\mathcal{W\,Q\,C\,M}$ | 57.6 | 33.9 | 38.4 | 62.1 | -2.11 | $\mathcal{Q\,M\,W\,C}$ | 35.2 | 41.1 | 26.1 | 30.8 | 21.16 |

## A.3 DETAILED HYPER-PARAMETERS

We summarize the hyperparameters used in our experiments in Table 5. These values were chosen through a combination of prior work, small-scale ablations, and practical compute considerations.

Table 5: Training Hyperparameters for All Training Settings. **ST** = Single-Task RL (e.g., **ST Math =** RL trained only on math).

| Hyperparameter | Curr. Learning | Joint Training | ST Coding | ST Math | ST QA | ST Writing | SFT |
|---|---|---|---|---|---|---|---|
| *Model Configuration* | | | | | | | |
| Max Prompt Length | 1024 | 1024 | 1024 | 1024 | 1024 | 1024 | - |
| Max Response Length | 3072 | 3072 | 3072 | 3072 | 3072 | 3072 | - |
| *Training Settings* | | | | | | | |
| Train Batch Size | 256×6 | 256×6 | 256×6 | 256×6 | 256×6 | 256×6 | 128 |
| Learning Rate | 1e-6 | 1e-6 | 1e-6 | 1e-6 | 1e-6 | 1e-6 | 2.5e-6 |
| Learning Scheduler | Constant | Constant | Constant | Constant | Constant | Constant | Cosine |
| Optimizer | AdamW | AdamW | AdamW | AdamW | AdamW | AdamW | AdamW |
| Grad Clip | 1.0 | 1.0 | 1.0 | 1.0 | 1.0 | 1.0 | 1.0 |
| Max Epoch | 3 | 3 | 3 | 3 | 3 | 3 | 3 |
| *RL Settings* | | | | | | | |
| KL Beta | 0.0 | 0.0 | 0.0 | 0.0 | 0.0 | 0.0 | - |
| Clip Ratio Low | 0.2 | 0.2 | 0.2 | 0.2 | 0.2 | 0.2 | - |
| Clip Ratio High | 0.2 | 0.2 | 0.2 | 0.2 | 0.2 | 0.2 | - |
| $N$ Rollouts | 16 | 16 | 16 | 16 | 16 | 16 | - |
| Rollout Temperature | 1.0 | 1.0 | 1.0 | 1.0 | 1.0 | 1.0 | - |
| Rollout Top-P | 1.0 | 1.0 | 1.0 | 1.0 | 1.0 | 1.0 | - |
| Rollout Top-K | 50 | 50 | 50 | 50 | 50 | 50 | - |
| *LLM-as-a-Judge Settings* | | | | | | | |
| Judge Model | gpt-4.1-mini | gpt-4.1-mini | - | - | - | gpt-4.1-mini | - |
| Reference Model | Qwen2.5-7B-Instruct | Qwen2.5-7B-Instruct | - | - | - | Qwen2.5-7B-Instruct | - |
| Temperature | 0.4 | 0.4 | - | - | - | 0.4 | - |

## A.4 CREATIVE WRITING TASK ABLATION

Table 6: Creative Writing performance across training methods, evaluated using different reference models. This comparison shows that the relative performance ordering of the training methods remains consistent across reference models. **ST** = Single-Task RL (e.g., **ST Math** trains only on the math task). **MM** = Model Merging. **JT** = Joint Training. **CL** = Curriculum Learning.

| Reference Model | ST Coding | ST Math | ST QA | ST Writing | SFT | MM | JT | $CL_{best}$ |
|---|---|---|---|---|---|---|---|---|
| GPT-4 | 71.6 (5th) | 74.2 (4th) | 63.0 (7th) | 78.3 (3rd) | 44.2 (8th) | 67.5 (6th) | 83.0 (2nd) | **84.2 (1st)** |
| GPT-4.1 mini | 35.0 (5th) | 48.3 (4th) | 30.8 (7th) | 55.0 (2nd) | 15.8 (8th) | 33.3 (6th) | 55.0 (2nd) | **61.6 (1st)** |
| Qwen2.5-7B Instruct | 40.0 (5th) | 46.6 (4th) | 32.5 (7th) | 65.8 (3rd) | 17.5 (8th) | 40.0 (5th) | 75.0 (2nd) | **75.0 (1st)** |

Table 7: Creative Writing performance across curriculum schedulings. Evaluating with different reference models GPT-4 GPT-4.1-mini and Qwen2.5-7B-Instruct), yields the same ordering of curricula once the variability implied by the standard deviations is taken into account. This demonstrates that the benefit of our BWT-guided curriculum design is robust across reference models and is not due to a particular of reference style.

| Curriculum | GPT-4 | GPT-4.1 mini | Qwen2.5-7B Instruct |
|---|---|---|---|
| $\mathcal{CMQW}$ | $84 \pm 2.9$ | $62 \pm 3.8$ | $75 \pm 3.4$ |
| $\mathcal{QMWC}$ | $79 \pm 3.2$ | $57 \pm 3.9$ | $68 \pm 3.7$ |
| $\mathcal{QWCM}$ | $82 \pm 3.0$ | $65 \pm 3.8$ | $74 \pm 3.5$ |
| $\mathcal{WQMC}$ | $75 \pm 3.4$ | $37 \pm 3.8$ | $46 \pm 3.9$ |

## A.5 DIVERSITY ANALYSIS FOR CREATIVE WRITING

To examine whether our creative writing task reward induces stylistic bias toward the reference responses, we conducted a diversity analysis using the MT-Bench evaluation set. Each MT-Bench example is a multi-turn conversational prompt; for every conversation, we computed ROUGE-L similarity between the model's generated turns and the corresponding reference turns, averaged the similarity over all turns within that conversation, and then averaged across the entire dataset. This yields a corpus-level average similarity metric, where lower similarity indicates greater stylistic diversity relative to the reference, a diversity measure used in Wang et al. (2023). Using this metric, we compared the base model and the model after curriculum-based RL fine-tuning. The average similarity decreased from 0.2226 (base) to 0.1864 (after Curriculum RL), suggesting that the model's writing becomes more stylistically diverse rather than converging toward the reference responses. This confirms that our hybrid reward and curriculum framework does not induce style collapse and that improvements on creative writing are not driven by imitation of a specific reference style.

## A.6   PROOF OF THEOREM 1

**Theorem 2.** *Under Assumptions 1 and 2, for any curriculum order $\sigma \in \mathfrak{S}_K$ starting from $\theta_0$, the terminal accuracies $\{\mathrm{Acc}(\theta_{s_{\max}}, T_k)\}_{k=1}^K$ given the initialization accuracies $\{\mathrm{Acc}(\theta_0, T_k)\}_{k=1}^K$, the $\mathrm{BWT}(\theta_0)$ and a curriculum $\sigma$ is exactly the output of Algorithm 1*

*Proof.* Consider a curriculum $(T_{\sigma(1)}, \cdots, T_{\sigma(K)})$ and the corresponding sequence of weights $\theta_0, \cdots, \theta_K$, ie $\theta_{t+1} = \theta(\theta_i, T_{\sigma(t+1)})$. Let denote by $\mathrm{Acc}(\theta_i)$ the accuracy array $(\mathrm{Acc}(\theta_t, T_j))_{j=1}^K$ By definition of the BWT matrix, for every $t \in \{0, \cdots, K-1\}$:

$$\log \mathrm{Acc}(\theta_{t+1}) = \left(\mathrm{BWT}(\theta_t)_{i\sigma(t+1)}\right)_{i=1}^K + \log \mathrm{Acc}(\theta_t) \tag{15}$$

.

By assumption 1, for all $i \neq \sigma(t+1)$:

$$\mathrm{BWT}(\theta_t)_{i\sigma(t+1)} = \mathrm{BWT}(\theta_0)_{i\sigma(t+1)}. \tag{16}$$

By assumption 2,

$$\begin{aligned}
\mathrm{BWT}(\theta_t)_{\sigma(t+1)\sigma(t+1)} &= \log \mathrm{Acc}(\theta_{t+1}, T_{\sigma(t+1)}) - \log \mathrm{Acc}(\theta_t, T_{\sigma(t+1)}) \tag{17} \\
&= \log \mathrm{Acc}(\theta_0, T_{\sigma(t+1)}) - \log \mathrm{Acc}(\theta_t, T_{\sigma(t+1)}). \tag{18}
\end{aligned}$$

As a result, by taking exponential of the equality above and by induction, for each $i \in \{1, \cdots, K\}$, the variable $a_i$ after step $j$ of the outer for loop of algorithm 1 contains $\mathrm{Acc}(\theta_{t=j}, T_i)$. When the algorithm stops, $j = K$ and we have $\theta_{t=K} = \theta_{t=s_{\max}}$ so that the algorithm outputs $a_i = \mathrm{Acc}(\theta_{s_{\max}}, T_i)$.

$\square$

## A.7   MODEL SIZE SCALING ABLATION

To examine whether our multi-task RL findings hold across model size, we additionally train a substantially smaller model, Qwen2.5-1.5B-Instruct, using the same hyperparameters and prompts as in the 7B experiments to ensure a clean and fair scaling comparison. The results in Table 8 reveal that while the 7B model reliably benefits from RL across all domains, the 1.5B model benefits less from single-task RL and notably suffers a drop in General QA performance. Qualitative inspection shows that the smaller model tends to exploit the output-format reward rather than engage in genuine chain-of-thought reasoning, generating syntactically valid `<think>` blocks with little substantive content. Coding gains are also limited, suggesting difficulty in producing correct solutions during rollouts. Because the model struggles to improve in isolation, curriculum effects become uninformative at this scale.

Additionally, we obtained consistent findings when repeating the experiment on two 1.5B models (DeepSeek-R1-Distill-Qwen-1.5B and Qwen3-1.5B), both of which exhibited the same reward-hacking behavior on General QA.

Overall, significantly smaller models fail to produce reliable solutions during RL rollouts, leading to negligible gains. This indicates that current models at this scale lack the foundational capacity necessary for meaningful multi-task training.

Table 8: Model size scaling ablation on small model, Qwen2.5-1.5B-Instruct, across four domains. **ST** = Single-Task RL. Bold = best; underline = second-best per row.

| Eval Task | Base Model | ST Coding | ST Math | ST QA | ST Writing |
|---|---|---|---|---|---|
| **Math** | | | | | |
| AIME24 | 0.7 | 2.7 | **7.3** | 4.7 | 4.0 |
| AMC23 | 27.5 | 32.0 | **33.5** | **33.5** | 28.5 |
| Gaokao2023en | 52.5 | 53.3 | 52.2 | 53.3 | **53.5** |
| MATH500 | 54.2 | 52.8 | **57.4** | 56.8 | 55.0 |
| MinervaMath | 29.8 | 31.6 | 30.2 | 28.6 | **34.2** |
| OlympiadBench | 23.1 | 21.5 | **24.4** | 22.1 | 23.7 |
| **Average** | 31.3 | 32.3 | **34.2** | 33.1 | 33.2 |
| **General QA** | | | | | |
| Biology | 26.6 | 19.9 | 32.1 | 11.9 | **32.4** |
| Business | 20.9 | 17.0 | 19.5 | 11.5 | **24.2** |
| Chemistry | 15.9 | 13.1 | 15.4 | 10.3 | **17.0** |
| CS | 21.2 | 20.5 | 26.1 | 10.5 | **28.3** |
| Economics | **26.9** | 17.9 | 26.8 | 12.7 | 26.3 |
| Engineering | **13.7** | 12.1 | 11.9 | 9.4 | 13.1 |
| Health | 16.3 | 13.2 | **22.6** | 11.7 | 17.9 |
| History | 15.0 | 12.9 | **17.1** | 9.7 | 14.7 |
| Law | **14.4** | 13.2 | 14.3 | 13.2 | 13.6 |
| Math | 33.1 | 27.5 | 35.4 | 11.7 | **37.8** |
| Other | 16.8 | 13.5 | **21.4** | 10.4 | 18.1 |
| Philosophy | 16.4 | 17.8 | **22.9** | 14.4 | 19.4 |
| Physics | 16.4 | 13.9 | 15.1 | 10.1 | **17.2** |
| Psychology | 21.9 | 22.3 | **29.7** | 11.9 | 26.1 |
| **Average** | 20.0 | 16.8 | **21.9** | 11.3 | 21.9 |
| **Coding** | | | | | |
| BigCodeBench | 19.7 | **21.5** | 20.3 | 19.7 | 20.8 |
| LiveCodeBench | 4.8 | **6.7** | 5.1 | 4.8 | 4.3 |
| **Average** | 12.3 | **14.1** | 12.7 | 12.3 | 12.5 |
| **Creative Writing** | | | | | |
| MT-Bench (Writing) | 10.8 | 14.2 | 20.8 | 11.7 | **43.3** |

