# OpenReview forum: "Omni-Thinker: Scaling Multi-Task RL in LLMs with Hybrid Reward and Task Scheduling"
_ICLR.cc/2026/Conference — Submitted to ICLR 2026_

### Official Review · Reviewer_qhMJ · 2025-10-27

**Soundness:** 2
**Presentation:** 3
**Contribution:** 2
**Rating:** 4
**Confidence:** 4

**Summary:**

This paper presents Omni-Thinker, a unified reinforcement learning framework for training large language models across different task domains. The approach combines hybrid reward signals, integrating rule-based verifiable rewards with LLM-as-a-Judge preference evaluations, and employs a backward-transfer guided curriculum scheduler to order tasks and reduce catastrophic forgetting. The authors claim improvements over joint training and model merging, and propose that simple assumptions on accuracy transfer can predict curriculum outcomes, with entropy dynamics explaining deviations in generative tasks.

**Strengths:**

- The use of backward transfer matrices to guide curriculum ordering is principled and builds on established continual learning concepts.
- The paper is well written and easy to follow.

**Weaknesses:**

Please see my detailed questions and concerns below.

**Questions:**

- Assumption 2 requires that seeing the full dataset once saturates task accuracy to the same level as training from initialization. How realistic is this for complex reasoning tasks? What happens when tasks require multiple epochs to converge?
- Theorem 1 is stated without proof. A statement without proof should not be called a "Theorem" in mathematics.
- For creative writing, you compare against a reference response from the dataset. Doesn't this bias the reward toward a specific style rather than encouraging diversity or creativity?
- The paper claims that models "learn to emulate lower or higher temperatures" (lines 384-387). Are there any direct evidence for this mechanistic claim beyond just correlational observations?
- Table 2 shows temperature ablations for only QA and Writing. Why not include Math and Coding?
- The creative writing evaluation uses MT-Bench against GPT-4 from 2023. This is now quite outdated. How would results change against more recent models?
- The proposed method requires computing the full BWT matrix upfront, which means training on all task pairs. For K tasks, it seems to require O(K^2) training runs. How does this scale computationally as K increases?
- The paper focuses on post-training of already instruction-tuned models. Would your approach work for continued pretraining or for training from scratch?

---

> ### Author Response · Authors · 2025-11-23
>
> We thank the reviewer for the detailed feedback. Below, we provide our responses to address the questions.
>
> **Q1: "Assumption 2 requires that seeing the full dataset once saturates task accuracy to the same level as training from initialization. How realistic is this for complex reasoning tasks? What happens when tasks require multiple epochs to converge?"**
>
> First, we would like to clarify that Assumption 2 is a recoverability assumption rather than a statement about convergence rate; it does not claim that a single epoch suffices for a model to reach its maximal accuracy. Rather, the assumption concerns recoverability: even if training on task $B$ temporarily reduces performance on task $A$, subsequent training on task $A$ is assumed to recover the same final accuracy one would obtain had task $A$ been trained in isolation. In other words, Assumption 2 states that any interference from other tasks is fully reversible through continued training on the affected task.
>
> Second, we observe that Creative Writing decreases the training effectiveness of subsequent tasks, whereas Coding increases it. We attribute this asymmetric behavior partly to an entropy-transfer effect, where tasks induce systematic shifts in the model’s output entropy that modulate optimization dynamics. This phenomenon directly illustrates a limitation of Assumption 2.
>
> Third, knowledge transfer between tasks can also break Assumption 2. Tasks such as Math and Coding share semantic and algorithmic structure, and accordingly we observe positive transfer between them. This beneficial interaction, while desirable, means that performance on a task can improve beyond what training in isolation would achieve.
>
> Fourth, we emphasize that Assumptions 1 and 2 are not intended to be literally true, but rather to initiate a hypothetico–deductive analysis that helps disentangle the mechanisms driving curriculum effectiveness. These simplifying assumptions allow us to derive quantitative predictions about curriculum performance and then measure how accurately these predictions match empirical outcomes. The discrepancies such as those caused by entropy transfer or knowledge transfer are precisely what reveal which factors most strongly influence curriculum success.
>
> Finally, we note that all tasks in our setting require multiple epochs of GRPO fine-tuning to converge, and continued training consistently improves accuracy. However, this does not affect the conceptual role of the assumptions, which concern recoverability and transfer structure rather than literal one-epoch convergence.
>
> **Q2: "Theorem 1 is stated without proof. A statement without proof should not be called a "Theorem" in mathematics."**
>
> We thank the reviewer for pointing this out. In the revised version, we have added the complete proof of Theorem 1 in Appendix A.6. We apologize for the earlier omission.
>
> **Q3: "For creative writing, you compare against a reference response from the dataset. Doesn't this bias the reward toward a specific style rather than encouraging diversity or creativity?"**
>
> To address the reviewer’s concern that using reference responses might bias the reward toward a specific style, we compute a diversity metric by measuring the average ROUGE-L similarity between model outputs and the reference answers before and after curriculum training. A reduction in similarity would indicate less stylistic imitation. We found that the average similarity decreased from 0.2226 to 0.1864, suggesting that the model’s writing style became more diverse relative to the reference rather than more aligned with it. This supports our claim that the hybrid RL and curriculum schedule do not cause the model to collapse toward a single reference style. The improvements in Creative Writing stem from better general generation quality rather than "bias the reward toward a specific style".
>
> Furthermore, we added evaluations against the more recent gpt-4.1-mini (2025-04-14) and our base model Qwen2.5-7b-instruct, providing the results in appendix A.4 Creative Writing Task Ablation. The Creative Writing performance ranking remained mostly the same across training methods and curricula for reference answers generated by different models. This suggests that our RL + curriculum improvements are not tied to a single reference model or a particular writing style.
>
> Nonetheless, we would like to clarify that the main goal of our work is understanding multi-task RL dynamics, hybrid reward integration, and backward-transfer-guided scheduling. Exploring task specific evaluation for creativity or diversity is an important direction, but it is not the focus of this work and our multi-task study.

---

> > ### Author Response · Authors · 2025-11-23
> >
> > **Q4: "The paper claims that models "learn to emulate lower or higher temperatures" (lines 384-387). Are there any direct evidence for this mechanistic claim beyond just correlational observations?"**
> >
> >
> > First of all, we apologize for the poor turn of phrase. Our intention was to suggest a possible mechanism by which one could have transfer of entropy without knowledge transfer.
> >
> > Second, we verified empirically that simple last-layer scaling is not responsible for the entropy shifts: comparing the norms and angles of the linear head before and after fine-tuning shows no evidence of a scalar rescaling consistent with temperature adjustment. However, beyond ruling out this trivial explanation, we do not yet have a definitive characterization of the mechanism that drives entropy changes across tasks.
> >
> > To avoid implying more certainty than our analysis supports, we have removed the speculative sentences in the revised version and now describe entropy transfer solely as an observed phenomenon, without attributing a specific underlying mechanism. We continue to investigate this behavior to find the underlying mechanism.
> >
> > **Q5: "Table 2 shows temperature ablations for only QA and Writing. Why not include Math and Coding?"**
> >
> > As shown empirically in Table 3, the gap between the best curriculum ($\mathcal{C}\mathcal{M}\mathcal{Q}\mathcal{W}$) and the worst curriculum ($\mathcal{W}\mathcal{Q}\mathcal{M}\mathcal{C}$) (curricula ranked by final geometrical average relative improvement detailed in Table 4) results in only minor drops for Math and Coding (around 3 percentage points), whereas QA and Writing experience much larger degradations (around 30 and 9 percentage points, respectively). Our goal in Table 2 was therefore to explain the cause of this discrepancy.
> >
> > The observed differences align with the entropy changes in Figure 2: open-ended tasks tend to increase entropy after RL fine-tuning, while structured/verifiable tasks tend to decrease it. Hence, the degree of “open-endedness" can be characterized by the entropy change: positive shifts imply more randomness in the outputs, whereas negative shifts signal a move toward deterministic outputs.
> >
> > Math and Coding rely on verifiable, deterministic reward signals, and RL on these tasks typically converges to low-entropy, near-deterministic outputs. Prior works, such as [1] [2], similarly reported entropy collapse in verifiable RL settings. For these reasons, temperature ablations offer limited additional insight for these two tasks.
> >
> > **Q6: "The creative writing evaluation uses MT-Bench against GPT-4 from 2023. This is now quite outdated. How would results change against more recent models?"**
> >
> > While GPT-4 may now be somewhat outdated as an reference model, our use of it follows the default MT-Bench evaluation rather than a choice specific to our work. All models (ST, SFT, merging, joint training, curriculum) are evaluated under the same judge, so GPT-4’s responses serve only as a consistent reference anchor.
> >     Nonetheless, to address the reviewer’s concern, we additionally evaluated against more recent model (GPT-4.1-mini from 2025-04-14) and the base model itself (Qwen2.5-7B-Instruct) in appendix A.4 Creative Writing Task Ablation. The relative ranking of methods or curriculum schedules remained unchanged, showing that our conclusions are not sensitive to the specific choice of the model providing the reference responses.

---

> > > ### Author Response · Authors · 2025-11-23
> > >
> > > **Q7: "The proposed method requires computing the full BWT matrix upfront, which means training on all task pairs. For K tasks, it seems to require O(K^2) training runs. How does this scale computationally as K increases?"**
> > >
> > > A closely related question was raised in Reviewer Afu3’s Q1, and we provide the same clarification here for completeness:
> > >
> > > We thank the reviewer for raising this important concern. Computing the backward-transfer matrices (accuracy and entropy) requires one RL training per task and one evaluation per ordered pair of tasks. The dominant cost is training, not evaluation:
> > >
> > > - Training each task once has the same computational cost as one full curriculum run because a curriculum also trains on each task to saturation.  Therefore, the overhead of our method is equivalent to **≈ one curriculum attempts**.
> > > -   A simple alternative method consisting in trying as little as 3 curricula already costs more.
> > > - Exhaustively evaluating all curricula would require $K!$ full trainings.
> > >
> > > Evaluation is deemed negligible in comparison: estimating each of the $ K^2 $ BWT entries within accuracy $ \epsilon $ scales as $ O(1/\epsilon^2) $, and forward-pass evaluation costs at least twice less than RL training both in memory and computations.
> > >
> > > Formally:
> > > $$ \mathrm{Cost} \simeq  \mathrm{ForwardPass} \times \left(
> > > 2\frac{\mathrm{ForwardPass} + \mathrm{Backprop}}{\mathrm{ForwardPass}} \times \mathrm{GRPOGroupSize} \times \mathrm{DatasetSize} + \frac{K^2}{\epsilon^{2}}\right).
> > > $$
> > >
> > > The GRPO Group Size is 16 in our experiment.
> > >
> > > However, a limitation occurs if one increases the task granularity by dividing the training dataset into smaller sub-tasks, then $K^2$ may grow while the total training dataset size is constant. Leading to a non-negligible BWT estimation.
> > >
> > > Thus, our method provides a near-optimal curriculum estimate for the cost of two curriculum runs. We will add a short scalability discussion to the Limitations section.
> > >
> > >
> > >
> > >
> > > **Q8: "The paper focuses on post-training of already instruction-tuned models. Would your approach work for continued pretraining or for training from scratch?"**
> > >
> > > We appreciate the reviewer’s interest. Our work focuses on the stage where RL is typically applied (post instruction-fine-tuning), once the model already has stable instruction following and reasoning ability. Whether our approach extends to continued pretraining or training from scratch depends less on our method and more on the feasibility of applying RL in those regimes. Training from-scratch via RL remains extremely difficult, while recent works, such as [3,4], demonstrates RL-style objectives can be incorporated into continued pretraining. Adapting our BWT-guided scheduling and hybrid rewards into RL continue pretraining is conceptually possible but nontrivial, and we view it as
> > > promising future work rather than a claim of this paper.
> > >
> > > ----
> > > [1] Wang, Shenzhi, et al. "Beyond the 80/20 rule: High-entropy minority tokens drive effective reinforcement learning for llm reasoning." arXiv preprint arXiv:2506.01939 (2025).
> > >
> > > [2] Zheng, Tianyu, et al. "First return, entropy-eliciting explore." arXiv preprint arXiv:2507.07017 (2025).
> > >
> > > [3] Dong, Qingxiu, et al. "Reinforcement Pre-Training." arXiv preprint arXiv:2506.08007 (2025).
> > >
> > > [4] Hatamizadeh, Ali, et al. "RLP: Reinforcement as a Pretraining Objective." arXiv preprint arXiv:2510.01265 (2025).

---

### Official Review · Reviewer_Afu3 · 2025-10-31

**Soundness:** 3
**Presentation:** 3
**Contribution:** 2
**Rating:** 6
**Confidence:** 3

**Summary:**

This paper proposes Omni-Thinker, a unified RL framework that scales LLMs across diverse tasks using hybrid rewards and backward-transfer-guided scheduling. The method integrates verifiable rewards for structured tasks with preference-based LLM-as-a-Judge evaluations for open-ended tasks. Experiments across four domains show average gains of 6.2% over joint training and 12.4% over model merging. The proposed method offers an effective solution for unified multi-task learning in both structured and open-ended domains.

**Strengths:**

1. This framework addresses the inconsistency in optimization direction across different tasks in the reinforcement learning process, integrating verifiable rule-based rewards and preference-based LLM evaluation into a unified reinforcement learning paradigm.

2. The proposed BMT, by quantifying how learning a task influences the performance of previously learned tasks, provides a referable paradigm for the learning order in curriculum learning, mitigating the catastrophic forgetting problem to some extent.

3. In experiments across four different domains, the proposed method demonstrates stable performance improvements, outperforming existing approaches to model merging and joint training.

**Weaknesses:**

1. As mentioned in the article, the overhead of curriculum scheduling increases gradually with the increase in workload, and the scalability of the proposed method may be limited. Are there efficient strategies for real-world deployment?
2. The paper presents results using Qwen2.5-7B as the base model for all experiments. Would the same backward-transfer-guided scheduling strategy remain optimal for significantly smaller or larger models?
3. The overall framework, particularly the curriculum design and entropy analysis, appears somewhat heuristic and lacks tight integration with the core methodological contributions. To strengthen the contribution, could the insights from the entropy analysis be more formally integrated into the scheduling algorithm itself?

**Questions:**

See weaknesses.

---

> ### Author Response · Authors · 2025-11-23
>
> **W1: "As mentioned in the article, the overhead of curriculum scheduling increases gradually with the increase in workload, and the scalability of the proposed method may be limited. Are there efficient strategies for real-world deployment?"**
>
>
> We thank the reviewer for raising this important concern. Computing the backward-transfer matrices (accuracy and entropy) requires one RL training per task and one evaluation per ordered pair of tasks. The dominant cost is training, not evaluation:
>
> - Training each task once has the same computational cost as one full curriculum run because a curriculum also trains on each task to saturation.  Therefore, the overhead of our method is equivalent to **≈ one curriculum attempts**.
> -   A simple alternative method consisting in trying as little as 3 curricula already costs more.
> - Exhaustively evaluating all curricula would require $K!$ full trainings.
>
> Evaluation is deemed negligible in comparison: estimating each of the $ K^2 $ BWT entries within accuracy $ \epsilon $ scales as $ O(1/\epsilon^2) $, and forward-pass evaluation costs at least twice less than RL training both in memory and computations.
>
> Formally:
> $$ \mathrm{Cost} \simeq  \mathrm{ForwardPass} \times \left(
> 2\frac{\mathrm{ForwardPass} + \mathrm{Backprop}}{\mathrm{ForwardPass}} \times \mathrm{GRPOGroupSize} \times \mathrm{DatasetSize} + \frac{K^2}{\epsilon^{2}}\right).
> $$
> The GRPO Group Size is 16 in our experiment.
>
> However, a limitation occurs if one increases the task granularity by dividing the training dataset into smaller sub-tasks, then $K^2$ may grow while the total training dataset size is constant. Leading to a non-negligible BWT estimation.
>
> Thus, our method provides a near-optimal curriculum estimate for the cost of two curriculum runs. We will add a short scalability discussion to the Limitations section.
>
> **W2 "Would the same backward-transfer-guided scheduling strategy remain optimal for significantly smaller or larger models?"**
>
> We appreciate the reviewer’s interest in understanding the scalability of our scheduling strategy. Running curriculum-scheduling experiments on significantly larger models is prohibitively expensive due to the number of curriculum permutations that must be evaluated for the study, each requiring full RL training.
>
> Smaller base models are primarily optimized for efficiency and can demonstrate either specialist reasoning or broad generalist behavior, but typically cannot support both simultaneously. Their limited capacity makes it difficult for them to accommodate the diverse reasoning requirements of a multi-task RL setup in the post-training regime, making them unsuitable as a starting base model for our study.
>
> Nevertheless, we are conducting an additional set of experiments on a significantly smaller model to further verify the consistency of our findings, and we will include the updated results.
>
> **W3 "Could the insights from the entropy analysis be more formally integrated into the scheduling algorithm itself?"**
>
> We agree this is a promising direction. A straightforward extension would be to classify tasks as entropy-increasing or entropy-decreasing and restrict the search space to curricula that schedule:
>
> - entropy-decreasing tasks first,
> - entropy-increasing tasks last.
>
> However, we consider this beyond the scope of the current work for two reasons:
>
> 1. **Dataset coverage limitations.**
>    Only one task among our four (Creative Writing) is empirically entropy-increasing. A principled entropy-aware scheduler would require multiple such tasks. The reason why Creative Writing increases entropy is an empirical finding without a mechanistic explanation. Identifying additional entropy-increasing domains would require new dataset construction.
>
> 2. **Stability considerations.**
>    Strict entropy-based ordering interacts with known RL fine-tuning instabilities:
>    - Long training on entropy-decreasing tasks can cause model collapse,
>    - Mixing entropy-increasing and entropy-decreasing phases requires explicit entropy or temperature control,
>    - A principled integration requires disentangling accuracy transfer from entropy transfer.

---

> ### Author Response · Authors · 2025-11-26
>
> **Follow-up of W2:**
> As promised in our initial response, we conducted an additional set of experiments using Qwen2.5-1.5B-Instruct to verify the consistency of our findings on a significantly smaller model. The single-task results are summarized in Table below:
> ## Performance across benchmarks (Qwen2.5-1.5B-Instruct)
> **ST = Single-Task RL**
>
> ---
>
> ## **Math**
> | Eval Task       | Base | ST Coding | ST Math | ST QA | ST Writing |
> |-----------------|------|-----------|---------|-------|------------|
> | AIME24          | 0.7  | 2.7       | **7.3** | _4.7_ | 4.0        |
> | AMC23           | 27.5 | 32.0      | **33.5** | **33.5** | 28.5 |
> | Gaokao2023en    | 52.5 | 53.3      | 52.2    | 53.3  | **53.5**  |
> | MATH500         | 54.2 | 52.8      | **57.4** | _56.8_ | 55.0 |
> | MinervaMath     | 29.8 | 31.6      | 30.2    | 28.6  | **34.2**  |
> | OlympiadBench   | 23.1 | 21.5      | **24.4** | 22.1 | _23.7_ |
> | **Average**     | 31.3 | 32.3      | **34.2** | 33.1 | _33.2_ |
>
> ---
>
> ## **General QA**
> | Eval Task  | Base | ST Coding | ST Math | ST QA | ST Writing |
> |------------|------|-----------|---------|-------|------------|
> | Biology    | 26.6 | 19.9 | 32.1 | 11.9 | **32.4** |
> | Business   | 20.9 | 17.0 | 19.5 | 11.5 | **24.2** |
> | Chemistry  | 15.9 | 13.1 | 15.4 | 10.3 | **17.0** |
> | CS         | 21.2 | 20.5 | 26.1 | 10.5 | **28.3** |
> | Economics  | **26.9** | 17.9 | _26.8_ | 12.7 | 26.3 |
> | Engineering| **13.7** | 12.1 | 11.9 | 9.4 | _13.1_ |
> | Health     | 16.3 | 13.2 | **22.6** | 11.7 | _17.9_ |
> | History    | 15.0 | 12.9 | **17.1** | 9.7 | 14.7 |
> | Law        | **14.4** | 13.2 | _14.3_ | 13.2 | 13.6 |
> | Math       | 33.1 | 27.5 | 35.4 | 11.7 | **37.8** |
> | Other      | 16.8 | 13.5 | **21.4** | 10.4 | _18.1_ |
> | Philosophy | 16.4 | 17.8 | **22.9** | 14.4 | _19.4_ |
> | Physics    | 16.4 | 13.9 | 15.1 | 10.1 | **17.2** |
> | Psychology | 21.9 | 22.3 | **29.7** | 11.9 | _26.1_ |
> | **Average** | 20.0 | 16.8 | **21.9** | 11.3 | **21.9** |
>
> ---
>
> ## **Coding**
> | Eval Task       | Base | ST Coding | ST Math | ST QA | ST Writing |
> |-----------------|------|-----------|---------|-------|------------|
> | BigCodeBench    | 19.7 | **21.5** | 20.3 | 19.7 | _20.8_ |
> | LiveCodeBench   | 4.8  | **6.7** | _5.1_ | 4.8 | 4.3 |
> | **Average**     | 12.3 | **14.1** | _12.7_ | 12.3 | 12.5 |
>
> ---
>
> ## **Creative Writing**
> | Eval Task          | Base | ST Coding | ST Math | ST QA | ST Writing |
> |--------------------|------|-----------|---------|-------|------------|
> | MT-Bench (Writing) | 10.8 | 14.2      | _20.8_ | 11.7 | **43.3** |
>
> Unlike the 7B model, which consistently improves across all tasks whether trained independently or together, the 1.5B model exhibits a substantial deterioration on General QA (20.0\% $\rightarrow$ 11.3\% for ST-QA).  Our qualitative analysis of the 1.5B model's outputs revealed reward hacking. We incentivize Chain-of-Thought reasoning by enforcing a $\<think\>...\</think\>$ structure followed by $\<answer\>...\</answer\>$. While the 7B model utilized this structure to generate meaningful reasoning traces, the 1.5B model learned to "game" the format reward: it generated valid tags but filled the \<think\> block with trivial or short sentences, effectively skipping the reasoning process entirely. Improvements in Coding were marginal (12.3\% $\rightarrow$ 14.1\%), further indicating that the model struggles to optimize the RL objective effectively in this domain compared to the 7B baseline.
>
> Since the 1.5B model fails to improve on single-task RL, examining curriculum effects no longer yields meaningful conclusions in this setting. These findings support our view that models substantially smaller than 7B may not provide a sufficiently strong starting point for our multi-task RL study, as the base model must first be capable of learning each task in isolation for curriculum effects to be meaningfully assessed.

---

### Official Review · Reviewer_XPqG · 2025-11-03

**Soundness:** 2
**Presentation:** 3
**Contribution:** 2
**Rating:** 2
**Confidence:** 4

**Summary:**

This paper studies training curriculum for multi-task RL with both verifiable and preference-based tasks. It considers a multi-stage training setup, where the model is trained only on a single task at each stage and only enters the next stage after it finishes all training samples of this task. The proposed approach first measures the cross-task influence — the impact of training on each individual task on the performance of other tasks — and then determines an optimal task ordering by prioritizing the task that yields the highest average test accuracy across the remaining tasks. Experiments shows that this curriculum-based strategy outperforms no curriculum training, model merging, and SFT.

**Strengths:**

* This paper proposes a simple framework for ordering tasks. The final ordering heuristics make intuitive sense.

**Weaknesses:**

* This work relies on overly simplisitc assumptions (for both assumptions) and there are no sufficient evidence to justify them. Also see questions section.
* Authors claim that the predicted accuracy using test set backward transfers are surprisingly precise, however, table 3 shows relatively low correlations between test and predicted accuracies.

**Questions:**

If inter-task transfer effects are assumed to be constant (i.e., independent of starting ckpt), and we have task-wise saturation, then the cumulative effect of sequential training should be order-invariant. Could the authors explain this?

---

> ### Author Response · Authors · 2025-11-23
>
> We thank the reviewer for the questions. Below we provide clarifications addressing the points raised, particularly regarding the role and interpretation of our theoretical assumptions, the scope of our predictive model, and the order-dependence of cumulative transfer effects.
>
> **W1: "This work relies on overly simplisitc assumptions (for both assumptions) and there are no sufficient evidence to justify them. Also see questions section."**
>
> We agree that Assumption 1 (constant off-diagonal BWT) and Assumption 2 (task-wise saturation) are idealized. Our intention was not to claim that they hold in full generality, nor to defend them as realistic models of LLM behavior. Instead, we use them as a tractable analytical lens that allows computing coarse curriculum predictions a priori without training $ K! $ orders.
>
> To avoid misunderstanding, we have rewritten the subsection “Reasonableness and limitations” to explicitly state:
> - The assumptions deliberately ignore known drivers of transfer, such as entropy drift and domain transfer;
> - They are simplified approximations used to understand why BWT-based ordering often works despite imperfect predictions;
> - Entropy transfer empirically violates both assumptions, which we use to explain systematic deviations (e.g., Creative Writing consistently outperforming its predicted accuracy).
>
> This aligns with the reviewer’s concerns and clarifies the intended scope of the theoretical model.
>
> **W2: "Authors claim that the predicted accuracy using test set backward transfers are surprisingly precise, however, table 3 shows relatively low correlations between test and predicted accuracies."**
>
> We respectfully clarify our statement regarding the predictions:
>
> - The goal of the predictive model is not to obtain high-fidelity accuracy estimates, but to generate a coarse ranking of curricula.
> - Despite the simplifying assumptions, the predictions are more accurate than expected.
>     -  Due to Algorithm 1 combining 9 evaluations per curriculum, the 95% CI corresponds to ≈6σ rather than the usual 2σ.
>     -  In Table 3, approximately half of the test/prediction gaps fall within the 95% confidence interval.
>     - Several predictions (e.g., best curriculum: Code→Math→QA→Writing) match empirical outcomes extremely well.
> - When the predictions fail, they do so with systematic bias, not noise: Creative Writing is consistently underestimated. We identify entropy transfer as a plausible, measurable source of this discrepancy (Sections 5.2–5.3).
>
> We have clarified these points in the revised text: we do not claim high correlation, only that the approximation is surprisingly informative given its crudeness.
>
> **Q1: "If inter-task transfer effects are assumed to be constant (i.e., independent of starting ckpt), and we have task-wise saturation, then the cumulative effect of sequential training should be order-invariant. Could the authors explain this?"**
>
> We believe the reviewer’s concern stems from a subtle but crucial misunderstanding. The statement that the cumulative effect is order invariant is inaccurate, our Theorem 1 formalizes this.
>
> Under Assumptions 1 and 2:
> After training on task  $T$, the accuracy on  $T$  resets to its saturated value, $ \mathrm{Acc}(\theta(T),T) $ independent of prior history. However:
> The accuracy on other tasks $T_i$  at that moment is not saturated; It is modified by all subsequent tasks through the off-diagonal terms $\mathrm{BWT}_{ik}$.
> Thus: The final accuracy on each task is determined by the sequence of BWT effects from tasks trained after it.
> Since the future tasks depend on the order, the cumulative effect is order-dependent, not invariant.
>
> Hence, even with constant BWT and saturation, the final model is not order-invariant.
> Only if all off-diagonal BWT terms were zero would the final model be the same regardless of schedule—a trivial case excluded by our empirical BWT matrices.
>
> We have revised the text to make this reasoning more transparent.

---

### Meta-Review · Area_Chair_PV2U · 2025-12-15

**Summary:**

The overly-simplified assumptions, lack of experiments on larger or smaller models and limited core methodological contributions concern reviewers. Two reviewers vote for rejection, where one votes for strong rejection, while another votes for marginal accept. AC believes it is hard for the authors to totally change reviewers' opinion by the rebuttal and no one will be willing to champion the paper. Given this, AC would recommend rejection of this paper.

**Reviewer Concerns:**

Reviewer concerns AC thinks were addressed by the rebuttal:

1. Low correlations between test and predicted accuracies.

AC comment: AC feels comfortable after seeing "The goal of the predictive model is not to obtain high-fidelity accuracy estimates, but to generate a coarse ranking of curricula."

2. Theorem 1 is without a proof.

AC's comment: The proof was added by the authors during the rebuttal. However, this lowers the credibility of this paper.

Reviewer concerns AC thinks are still outstanding:

1. Assumptions 1 and 2 are overly simplified (this is a common concern raised by two reviewers).

AC's comment: The authors agreed that Assumption 1 (constant off-diagonal BWT) and Assumption 2 (task-wise saturation) are idealized. The intention was not to claim that they hold in full generality, nor to defend them as realistic models of LLM behavior. Therefore, the concerns are still outstanding. The authors claimed they will rewrite the section to clarify it. However, AC believes the concerns would remain.

2. Lack of experiments for significantly smaller or larger models.

AC's comment: The authors did not provide any additional experimental results in the rebuttal. So the concerns remain.

3. The proposed framework lacks tight integration with the core methodological contributions.

AC's comment: The authors did not provide a straightforward response to this comment. They only claimed that one direction that the reviewer proposed is promising.

4. "The paper claims that models "learn to emulate lower or higher temperatures" (lines 384-387). Are there any direct evidence for this mechanistic claim beyond just correlational observations?"

AC's comment: The paper seems to have a writing issue in the first round of review (though the authors claimed to remove the "speculative sentences" in the revision.).

**Reviewer Scores:**

AC believes each reviewer will have remaining concerns after the rebuttal. So with a high chance, their scores will not be changed. Given the scores in the first of review, AC would lean towards rejecting the paper.

---

### Decision · Program_Chairs · 2026-01-26

Reject